# nNOS Increases Fiber Type-Specific Angiogenesis in Skeletal Muscle of Mice in Response to Endurance Exercise

**DOI:** 10.3390/ijms24119341

**Published:** 2023-05-26

**Authors:** Oliver Baum, Felicitas A. M. Huber-Abel, Martin Flück

**Affiliations:** 1Institute of Physiology, Charité-Universitätsmedizin, 10117 Berlin, Germany; 2Institute of Anatomy, University of Bern, 3012 Bern, Switzerland; 3Heart Repair and Regeneration Laboratory, Department EMC, Faculty of Science and Medicine, University of Fribourg, 1700 Fribourg, Switzerland; martin.flueck@unifr.ch

**Keywords:** angiogenesis, capillary, endurance exercise, nNOS, nitric oxide, skeletal muscle

## Abstract

We studied the relationship between neuronal NO synthase (nNOS) expression and capillarity in the tibialis anterior (TA) muscle of mice subjected to treadmill training. The mRNA (+131%) and protein (+63%) levels of nNOS were higher (*p* ≤ 0.05) in the TA muscle of C57BL/6 mice undergoing treadmill training for 28 days than in those of littermates remaining sedentary, indicating an up-regulation of nNOS by endurance exercise. Both TA muscles of 16 C57BL/6 mice were subjected to gene electroporation with either the pIRES2-ZsGreen1 plasmid (control plasmid) or the pIRES2-ZsGreen1-nNOS gene-inserted plasmid (nNOS plasmid). Subsequently, one group of mice (*n* = 8) underwent treadmill training for seven days, while the second group of mice (*n* = 8) remained sedentary. At study end, 12–18% of TA muscle fibers expressed the fluorescent reporter gene ZsGreen1. Immunofluorescence for nNOS was 23% higher (*p* ≤ 0.05) in ZsGreen1-positive fibers than ZsGreen1-negative fibers from the nNOS-transfected TA muscle of mice subjected to treadmill training. Capillary contacts around myosin heavy-chain (MHC)-IIb immunoreactive fibers (14.2%; *p* ≤ 0.05) were only higher in ZsGreen1-positive fibers than ZsGreen1-negative fibers in the nNOS-plasmid-transfected TA muscles of trained mice. Our observations are in line with an angiogenic effect of quantitative increases in nNOS expression, specifically in type-IIb muscle fibers after treadmill training.

## 1. Introduction

Skeletal muscle is one of the few tissues in adult mammals that retain the potential to increase the number of capillaries in response to appropriate stimuli. The corresponding process of capillary proliferation is designated angiogenesis. Over the years, it has become evident that the realization of angiogenesis in skeletal muscle depends on a complex interaction between pro-angiogenic and anti-angiogenic factors, although it is still largely unknown how this interaction is coordinated, as previously reported [1,2,3,4,5]. 

Due to its chemical properties, nitric oxide (NO) seems to be an ideal candidate for coordinating the interaction of these molecules and their integration into signaling networks to realize angiogenesis: it is a highly reactive, radical gas that can bind to specific receptor molecules after diffusing inside cells (autocrine), but can also pass through the plasma membrane to trigger signaling activities in adjacent cells (paracrine). It is furthermore noteworthy that NO is generated through the catalytic activity of three intracellular NO synthases (NOS), termed nNOS, iNOS and eNOS, with specific expression patterns, regulation and kinetics [6]. In consequence, the local availability of NO in all tissues is precisely controlled even when the environmental settings within the tissues change (e.g., the hemodynamic and metabolic conditions in skeletal muscle).

In skeletal muscle fibers of rodent muscles, neuronal NOS (nNOS) is present at high levels in the sarcolemma [7] and is preferentially expressed in oxidative type-II (IIa and IId/x) fibers [8,9]. Many functions are ascribed to nNOS in skeletal muscle [10]. In an autocrine manner, nNOS-derived NO influences several reactions of the oxidative metabolism in skeletal muscle fibers: nNOS interacts with PGC-1alpha [11], a master regulator gene of mitochondrial biogenesis, is involved in the establishment of an intact mitochondrial phenotype [9,12,13] and exhibits a positive allosteric effect on phosphofructokinase-1 activity [14]. In addition, nNOS activity lowers contraction force [7] and aids in the uptake of glucose into skeletal muscle fibers during exercise [15,16]. The common consequence of this broad spectrum of functions is the increase in glucose availability in skeletal muscle fibers either by accelerating anaplerotic reactions or inhibiting carbohydrate oxidation. In addition, nNOS is associated with maintaining fiber integrity at the protein level [17] or linked depending on the methylation state of the nNOS gene [18]. Furthermore, the alteration of nNOS expression levels within skeletal muscle fibers is accompanied by a change in the availability of superoxide and, subsequently, hydrogen peroxide and/or peroxynitrite suggesting that the enzyme is involved in scavenging of reactive oxygen species (ROS), which leads to cell protection and/or indirect cell signaling [19,20]. 

Given the sarcolemmal localization of nNOS in close proximity to the microcirculation [8] and its alleged impact on blood flow [21,22,23], it seems possible that nNOS also functions as a signaling enzyme that is involved in the paracrine communication between skeletal muscle fibers and the surrounding capillaries. Such an integrative character of the nNOS/NO-system in skeletal muscle would be particularly relevant for the induction of angiogenesis, which meets the higher demand for oxygen and substrates during endurance exercise [24]. However, a direct influence of nNOS on angiogenesis in skeletal muscle has not yet been demonstrated.

It was therefore the primary aim of this project to assess whether nNOS has an impact on angiogenesis in skeletal muscle in response to endurance exercise, which is an established trigger for the induction of physiological angiogenesis in the skeletal muscles of humans [25] and rodents [26]. To address this issue, we applied different experimental set-ups with mice subjected to treadmill training for induction of angiogenesis in their limb skeletal muscles. Our analysis revealed that nNOS positively affects the capillarity around glycolytic (type-IIb) skeletal muscle fibers and thus impacts the angiogenic potential of these muscle fibers, which are the most responsive ones to endurance exercise in mice [26]. 

## 2. Results

### 2.1. Treadmill Training of nNOS-KO Mice and WT Littermates

For visualization of the capillaries, cross-sections of the TA muscles were subjected to alkaline phosphatase histochemistry (Figure 1A). If the CF ratio on the cross-sections of the entire TA muscle from nNOS-KO mice and WT littermates either being trained for 28 days or remaining sedentary was compared, only non-significant differences (*p* > 0.05) were ascertained. However, when we used serial sections subjected to succinate dehydrogenase histochemistry (SDH) to distinguish between the two parts of the TA muscle that differ in their metabolic character (Figure 1B), we found higher CF ratio values (*p* ≤ 0.05) in the glycolytic part of the post-exercise TA muscles than in this region of the pre-exercise TA muscles in both the nNOS-KO mice (17%) and their WT littermates (14%), as shown in Figure 1C. These findings underline that angiogenesis occurred specifically in the glycolytic portion of the TA muscle in both the WT and nNOS-KO mouse strains. Notably, the CF ratio in the oxidative TA muscle portion tended to be lower in the nNOS-KO than the WT mice prior to training (8%; *p* = 0.07) and was also lower after treadmill training (−16%; *p* ≤ 0.05). 

In addition, we subjected the nNOS-KO mice and their WT littermates to two performance tests before and after the 28 days of endurance training (Figure 1D,E). Under basal conditions prior to the training, the differences in incremental and endurance tests were only non-significant (8% and 13%; *p* > 0.05) between the nNOS-KO mice and the WT mice. In both exercise tests, the performance of both mouse strains was improved (*p* ≤ 0.05) after the treadmill training: in the incremental test, the nNOS-KO mice ran 138% longer and their WT counterparts 53% after than before the training, while the running time in the endurance test increased by 277% (nNOS-KO) and 189% (WT) after the training period. Interestingly, the nNOS-KO mice ran longer (*p* ≤ 0.05) than the WT mice in both performance tests after the treadmill training (incremental test: 43%; endurance test: 33%).

### 2.2. Treadmill Training of C57BL/6 Mice

To assess whether the nNOS expression in skeletal muscle is regulated by endurance exercise, we quantified nNOS levels in the TA muscles of C57BL/6 mice that underwent treadmill training or remained sedentary for 28 days. The mRNA and protein levels of nNOS in the TA muscle were 131% and 63% higher (*p* ≤ 0.05), respectively, in the treadmill-trained mice than in their sedentary littermates (Figure 2). These findings indicate that the expression of nNOS in murine TA muscles is up-regulated in response to endurance exercise. 

### 2.3. Treadmill Training of WT Littermates Subjected to nNOS-Gene Electroporation

For the gene electroporation experiments, we used the expression plasmid pIRES2-ZsGreen1, which contains a multi-cloning site into which the nNOS gene was inserted and separated from the ZsGreen1 reporter gene by an IRES sequence (Figure 3A). This construction of the plasmid ensures that the muscle fibers expressing recombinant nNOS encoded by the plasmid also produce the ZsGreen1 protein. Either the control plasmid lacking a specific gene insert or the plasmid containing the nNOS gene was injected into one of the two TA muscles, which were then subjected to gene electroporation. Half of the mouse population was then exercised on the treadmill for seven days, while the other half remained sedentary. As a result, four groups of TA muscles were available at the end of the experiments: plasmid-transfected + sedentary, nNOS-transfected + sedentary, plasmid-transfected + trained and nNOS-transfected + trained.

Viewed from the muscle surface, many fibers of the TA muscles isolated eight days after the gene electroporation were found to be ZSgreen1-reactive (Figure 3B). The transfection efficiency being achieved by gene electroporation was quantified by fluorescence microscopy of cross-sections showing between 12 and 17% (*p* > 0.05) of the skeletal muscle fibers in the TA muscles of the four experimental groups to express the fluorescent reporter protein ZsGreen1 (Figure 3C,D). Immunohistochemistry using an anti-nNOS antibody in combination with green fluorescence microscopy confirmed the exclusive sarcolemmal expression of nNOS in both ZsGreen1-positive and ZsGreen1-negative muscle fibers (Figure 3E). Densitometric quantification of nNOS immunoreactivity in randomly selected muscle fibers revealed that gene electroporation with the nNOS plasmid resulted in higher nNOS expression by 18% (*p* ≤ 0.05) without and 23% (*p* ≤ 0.05) with treadmill training, as shown in Figure 3F.

To monitor whether the nNOS-gene transfer altered nNOS availability, we assessed and compared its NOS activity and nNOS expression levels in total TA muscles of the sedentary and the one-week-trained mice by multiple approaches (Figure 4). Quantitative evaluation of formazan production in the TA muscle cryosections revealed that nNOS-specific diaphorase activity did not differ significantly (18–27% deviation from control, *p* > 0.05) between the four study groups (Figure 4A,B). The nitrite production rate, which corresponds to NOS activity, in the TA muscle homogenates transfected with either the control plasmid or the nNOS plasmid was only non-significantly different (15%; *p* > 0.05) in the sedentary mice (Figure 4C). 

Real-time PCR showed that the nNOS mRNA expression levels were only non-significantly different between the study groups (17–45% deviation from control; *p* > 0.05) when the TA muscle homogenates transfected either with the plasmid or the nNOS plasmid combined with/without eight days of treadmill training were investigated (Figure 4D). In immunoblots on TA muscle extracts from mice of the four study groups, nNOS was detected as a double band at about 160-kDa (Figure 4E; Appendix A). Densitometric quantification of the nNOS bands showed (Figure 4F) that 87% higher (*p* ≤ 0.05) levels of nNOS were present in the plasmid-transfected TA muscle of mice subjected to one-week treadmill training than in the TA muscles of the sedentary mice. In contrast, transfection with the nNOS plasmid combined without/with training was accompanied by only non-significantly higher nNOS protein levels (19–44% deviation from control; *p* > 0.05). 

Since a significant increase of nNOS expression in total muscle homogenates in response to nNOS transfection was not detected, we continued our investigation at the muscle fiber level. To evaluate whether nNOS has an effect on capillarity, we performed immunohistochemistry using a polyclonal anti-ZsGreen1 antibody (to enhance the ZsGreen1 fluorescence signal) in combination with one of three monoclonal anti-myosin heavy-chain (MHC) antibodies (MHC-IIa; MHC-IId/x; MHC-IIb), as well as Bandeiraea simplicifolia (BS)-1 lectin histochemistry specific for capillaries in mouse skeletal muscle (Figure 5A,C,E; Appendix A). The fiber-type-specific quantification of the number of capillary contacts around individual skeletal muscle fibers (CC) in the four study groups revealed that the CC differed only non-significantly (*p* > 0.05) between ZsGreen1-positive and ZsGreen1-negative MHC-IId/x or MHC-IIa muscle fibers in all study groups (Figure 5B,D). In contrast, CC was 14.2% higher (*p* ≤ 0.05) around ZsGreen1-positive than ZsGreen1-negative MHC-IIb muscle fibers of nNOS-transfected TA muscle from mice undergoing treadmill training (Figure 5F). 

## 3. Discussion

The main aim of this project was to test the hypothesis that nNOS has an impact on angiogenesis in skeletal muscle. To address this issue, we carried out three connected series of experiments/studies with mice, which were subjected to treadmill training in order to induce proliferation of the capillary bed in their tibialis anterior (TA) muscle. Specifically, we found (1) higher increases in CF ratio within the glycolytic portion of the TA muscle after 28 days of treadmill training in nNOS-KO mice than in their WT littermates, (2) more nNOS expression at mRNA and protein levels in the TA muscle of C57BL/6 mice undergoing 28 days of treadmill training compared to their sedentary littermates, and (3) more capillary contacts around MHC type-IIb muscle fibers in the TA muscles of C57BL/6 mice subjected to gene electroporation with an nNOS plasmid in combination with seven days of treadmill training than in the control groups (transfection with the plasmid lacking a gene insert and/or remaining sedentary).

For quantification of capillarity in skeletal muscle, we used the numerical capillary-to-fiber ratio (CF ratio), which represents the number of capillaries divided by the number of skeletal muscle fibers within a given area [3]. This index should not be confused with the capillary density, which is the number of capillary profiles per cross-sectional area of muscle fibers. In contrast to the CF ratio, the CD depends on the mean cross-sectional area of muscle fibers, which is often modulated in response to changes in physical demand or the metabolic environment [3]. If the CF ratio in skeletal muscle changes over time, angiogenesis has occurred, and the magnitude of the change in CF ratio provides information about the extent of the angiogenic growth of the capillary system [3]. To date, most studies on angiogenesis in mouse skeletal muscle in response to exercise have been conducted by analyzing the plantaris and the gastrocnemius muscles, which are heavily strained by treadmill training due to their merging into the Achilles tendon. We chose to study the TA muscle for two reasons: (1) Since our aim was to characterize the influence of nNOS on angiogenesis in a mice muscle subjected to gene electroporation, the muscle to be analyzed had to be easily accessible for experimental manipulation by electrodes, i.e., it had to be located superficially in the lower limb, and (2) because we wanted to investigate the influence of endurance exercise on both glycolytic and oxidative fibers, which react differently to training [26], the mixed TA muscle seemed to be particularly suitable to us due to its even loading in dorsiflexion when running on a sloping treadmill. However, we would like to emphasize that a systematic evaluation of the angiogenic potential of the various skeletal muscles in the mouse lower leg is still pending. Such an analysis could lead to the identification of other skeletal muscles that meet the previously mentioned two conditions as well or even better than the TA muscle.

The nNOS-KO mice showed a higher incidence of angiogenesis in the glycolytic portion of the TA muscle and better performance than their WT littermates after the training period of 28 days. At first glance, these observations imply that the expression of nNOS inhibits the expansion of the capillary system in the TA muscle and prevents performance enhancement of mice in response to exercise. However, the findings can also be interpreted to mean that the nNOS-KO mice are structurally better equipped than WT mice to adapt more quickly and more efficiently to an exercise stimulus, e.g., by having a higher mitochondrial volume density in their skeletal muscles under basal conditions. In fact, several previous publications have quantified the mitochondria content in skeletal muscles from nNOS-KO mice and WT mice with conflicting results [9,12,13,27]. Based on these conflicting experimental data sets, we agree with previous conclusions [28] that the nNOS-KO mouse is not a suitable animal model for studies on skeletal muscle physiology due to multiple adaptations of these mice to the systemic lack of nNOS expression. Consequently, we prefer to neglect the findings gained with this animal model in the interpretation of exercise studies.

Several studies report that levels of NO [29] and nNOS [30,31,32] are up-regulated in the skeletal muscles of rodents after endurance exercise due to higher muscle fiber contractility or hypoxia. Our finding (higher nNOS levels in the TA muscle of C57BL/6 mice before than after the training period) is consistent with these previous investigations. All studies consistently suggest that endurance exercise triggers the up-regulation of nNOS expression in skeletal muscles, leading to higher availability of the signaling molecule NO within muscle fibers after a repetitive training stimulus.

Previous studies have shown that the stability of the transient expression plasmid decreases continuously after in vivo gene electroporation [33,34]. We therefore decided to train the mice on a treadmill for seven days after gene electroporation to then terminate the experiment by euthanizing the mice to remove their TA muscles. In fact, after this time we found a sufficient number of muscle fibers in the TA muscles transfected with the control plasmid or the nNOS plasmid (without/with exercise) to be ZsGreen1-positive (about 15% of all TA muscle fibers). 

Applying real-time PCR and quantitative immunoblotting, nNOS levels in the homogenates of the nNOS-transfected TA muscle did not differ significantly from those in plasmid-transfected TA muscles (without/with exercise). In addition, nitrate production rates were not higher in the nNOS-transfected TA muscle either. We hypothesize that the number of muscle fibers harboring the nNOS plasmid was too small in our experiment to facilitate substantiation of transfection-caused increases in nNOS expression in homogenates. Therefore, the assessment of the consequences of gene electroporation in our study has to be limited to the analysis at the individual fiber level.

The pIRES2-ZsGreen1 plasmid used to insert the nNOS gene by electroporation lacks promoter sequences (Figure 3A) to generate differential levels of nNOS expression in the set of transfected muscle fibers. However, skeletal muscle fibers still carry the intrinsic nNOS gene with regulatory promoter sequences, giving these fibers the potential to induce nNOS expression in response to endurance exercise. Accordingly, we were able to distinguish between skeletal muscles with four different nNOS expression patterns (corresponding to the assignment of the four mouse study groups): (1) intrinsic nNOS from sedentary muscle; (2) intrinsic nNOS + transfected nNOS from sedentary muscle; (3) intrinsic nNOS from trained muscle; (4) intrinsic nNOS + transfected nNOS from trained muscle. 

Since the IRES sequence was located 3′ downstream of the nNOS gene (with the initial AUG) but 5′ upstream of the ZsGreen1 reporter gene within the plasmid primary structure, expression of ZsGreen1 indicates transcription of the nNOS mRNA in full length. Actually, these ZsGreen1-positive muscle fibers contained significantly between 18 and 23% more nNOS immunoreactivity than the ZsGreen1-negative muscle fibers, confirming that the transformation not only increased ZsGreen1 production but also nNOS expression at the same time in susceptible muscle fibers of the murine TA muscles. In consequence, we confined the subsequent analysis to the comparison of ZsGreen1-positive with ZsGreen1-negative muscle fibers found on the same cryosections. After combining with fiber typing on serial sections, we found that only ZsGreen1-positive type-IIb fibers of the TA muscle had significantly more CCs than the ZsGreen1-negative type-IIb fibers after seven days of treadmill training, suggesting that angiogenesis was specifically accomplished around these muscle fibers in response to endurance exercise when the nNOS levels were elevated due to the experimental manipulation. 

Interestingly, the TA muscles of the mice that were transfected with the control plasmid and underwent treadmill training and the TA muscles of the sedentary mice transfected with the nNOS plasmid showed no significant differences in CC (even around type-IIb fibers). We explain this observation with the short span between gene electroporation and euthanasia of the mice, which requires the simultaneous effect of both up-regulated nNOS expression and treadmill training. 

Type-IIb muscle fibers have a much more extensive glycolytic metabolism, express lower levels of nNOS and are surrounded by fewer capillaries than the other type-II fibers [35]. Consequently, type-IIb muscle fibers have the greatest capacity to adapt to endurance exercise in rodents with a switch in metabolic profile [36] and an up-regulation of VEGF-A expression [37] related to higher capillarization [26]. We suggest that the up-regulation of nNOS by endurance exercise induces the onset of angiogenesis around type-IIb muscle fibers. This hypothesis is consistent with the finding that nNOS affects blood flow and vascular conductance, specifically in glycolytic skeletal muscles of rats undergoing high-intensity treadmill exercise [38]. 

Deduced from the results presented here and the literature, we propose a molecular model that explains how nNOS could affect exercise-induced angiogenesis: Endurance exercise increases muscle contractility and oxygen partial pressure, which leads to the up-regulation of nNOS expression and consequently higher NO availability in strained skeletal muscle fibers. Higher levels of nNOS-generated NO then induce the production and subsequent secretion of VEGF-A in an autocrine manner, particularly in type-IIb skeletal muscle fibers, or diffuse in a paracrine manner into adjacent capillary ECs to induce VEGF-A up-regulation. Finally, VEGF-A (irrespective of its cellular source) binds to its receptor VEGFR-2/KDR, which is expressed on the plasma membrane of capillary ECs, which then proliferate as a prerequisite for subsequent sprouting angiogenesis. 

We are aware that some methodological drawbacks may restrict the significance of our findings. (1) Skeletal muscles of the mouse hind-limb other than the TA muscle might be better suited to study the effect of nNOS on angiogenesis in response to exercise, but are not accessible to gene electroporation. (2) Mice in which the intact nNOS gene is systemically knocked-out do not represent a suitable model to study skeletal muscle angiogenesis due to compensatory adaptations [9] and would have to be replaced by other tools or methods (conditional, tissue-specific KO mice; nNOS-specific chemical inhibitors). (3) Since a gene electroporation-induced increase in nNOS expression was only detectable at the muscle fiber level, the molecular analysis of nNOS-mediated effects is methodologically problematic (e.g., to quantify VEGF-A concentrations). (4) The time period of 7 days of treadmill training after gene electroporation represents a compromise between the effect of endurance exercise and plasmid stability and may be too short for the observation of effects. 

In summary, we conclude that endurance-exercise-induced angiogenesis in skeletal muscle is influenced by NO generated by nNOS present in the sarcolemma of skeletal muscle fibers. The scientific literature suggests that nNOS expression might be increased due to the higher skeletal muscle contractility or hypoxia during training, whereas higher NO availability produced by eNOS might be triggered by higher shear stress in the capillaries. Thus, the influence of NO on VEGF-A expression depends on the localization site of the corresponding NOS form and the context of the physical stimulus, as previously suggested [39]. The study presented here provides additional information on the complex molecular interactions of nNOS leading to angiogenesis in skeletal muscle in response to endurance exercise.

## 4. Materials and Methods

### 4.1. Animals

Study 1: As previously reported [40], nine nNOS-KO mice and eight C57BL/6J wild-type (WT) littermates remaining sedentary, as well as four nNOS-knockout (KO) mice and four C57BL/6J WT mice subjected to treadmill training for 28 days, were included in this study. All mice were male and sacrificed at 16 weeks of age. The nNOS-KO mouse strain was originally generated by recombinant replacement of exon-2 of the nNOS gene with a neomycin cassette [41] and crossed into the C57BL/6J WT background for eight generations by our group [40]. 

Study 2: Sixteen male C57BL/6J at the age of 9 weeks were purchased from Charles River, Sulzfeld, Germany. Eight of these mice remained sedentary, while the other eight mice were subjected to treadmill training for 28 days, as previously reported [42]. All mice were sacrificed after the training period. 

Study 3: For the gene electroporation experiments, 16 male C57BL/6J mice (Charles River, Sulzfeld, Germany) at the age of six months were used. 

All mice were housed in a temperature-controlled room (21 °C) with a 12:12 h light–dark cycle. Animals were allowed chow pellets and water ad libitum. At sacrifice, mice were anesthetized with a ketamine/xylazine (100 mg × kg^−1^/5 mg × kg^−1^) cocktail via IP injection. The TA muscle of both legs was isolated, frozen in isopentane and subsequently liquid nitrogen to be stored at −80 °C until use. The animal protocols were approved by the Animal Protection Commission of Canton Bern, Switzerland (51/08 and 27/12).

### 4.2. Treadmill Training and Performance Tests

Four nNOS-KO mice and four WT littermates (Study 1), as well as eight C57/BL6J (Study 2), were subjected to a 28 days (=four weeks) period of forced treadmill exercise, consisting of six 45 min units per week. The training program was initiated at a speed of 16 m/min and a treadmill incline of 10°. The velocity and the incline were then incrementally increased on a weekly basis to maximum values of 25 m/min and 20°, respectively. The other nine nNOS-KO mice and eight C57BL/6J WT mice (Study 1), as well as four C57BL/6J WT mice (Study 2), remained sedentary to represent control groups. 

Two performance tests were performed before and after the last unit of the training period according to the following schedule: after the incremental test, there was a 2-day break before the endurance test was carried out. Two days prior to the incremental testing, the sedentary mice were familiarized with the treadmill (10 m/min for 10 min daily at a 5° incline). Both performance tests were discontinued when the animals were too exhausted to continue running despite the projection of air jets at the end of the treadmill. The results of both running performance tests were expressed as time to exhaustion in seconds.

The incremental tests were conducted at a fixed incline of the treadmill (10°) with a step-by-step increase in velocity. Initiated at 12 m/min, the speed was increased 4 m/min every 2 min. The endurance tests were carried out two days after the incremental test at a fixed incline (15°) and a steady treadmill speed, which was set individually for each mouse to 70% of the peak velocity determined in the incremental test. The mice were euthanized the day after the second endurance test.

### 4.3. Gene Electroporation 

The synthesis of the gene encoding nNOS alpha-isoform (4676 bp) was outsourced to Entelechon (Regensburg, Germany). This nucleotide sequence was subsequently inserted in the multicloning site of the transient expression plasmid pIRES2-ZsGreen1 (Clontech, Heidelberg, Germany) to obtain pIRES2-ZsGreen1-nNOS. Both plasmids were prepared and purified by PlasmidFactory (Bielefeld, Germany). 

Intramuscular gene transfer of the plasmid was achieved via injection of plasmid DNA and subsequent electric pulse delivery in both legs essentially as previously described [43]. Therefore, the mice were individually anesthetized with isoflurane. A 30 μg amount of expression plasmid in 30 μL of saline solution (0.9% NaCl) was injected with a sterile 0.3 mL syringe into both TA muscles (left TA: plasmid; right TA: nNOS plasmid) of the shaved lower limbs. After 5 min, electric pulses (3 strains of 100 pulses of 100 μs each at 50 mA) were delivered at two different locations in the central portion of the TA muscle using a GET42 pulser with needle electrodes (E.I.P. Electronique et Informatique du Pilat, Jonzieux, France). This technique typically results in over-expression of the plasmid for about one week [44]. The mice recovered rapidly from this procedure and began to move freely 30 min after the intervention. 

The mice were divided into two groups with eight (control mice that remained sedentary) and eight animals (exercising mice). The latter group was trained on a treadmill for 1 h per day at 16 m/min and with an incline increasing every second day from 9° to 15° for seven days beginning one day after the gene electroporation. Trained and sedentary mice were euthanized 24 h after the last training session and, thus, eight days after gene electroporation. The external view of the TA muscle was recorded through a fully automatic fluorescence stereomicroscope (Leica Microsystems M205FA, Wetzlar, Germany) equipped with the GFP Plus 480/40 nm filter and a mounted Canon EOS 5D Mark II camera.

### 4.4. RNA Extraction, Reverse Transcription and Real-Time PCR 

RNA extraction and reverse transcription using GoTaq Hot Start Polymerase (Promega, Dübendorf, Switzerland) were performed as previously described [45]. Real-time PCR was carried out using the ABI Prism 7900 HT sequence detection system and SYBR Green PCR Master Mix for quantitative PCR (Applied Biosystems, Rotkreuz, Switzerland), as previously described [45]. Primers were designed using the software Primer Express version 3.0.1 (Applied Biosystems, Waltham, MA, USA): nNOS exon 2 forward: 5′-CTT GGC TTG GAG GTC TTC TG-3′; nNOS exon 2 reverse: 5′-GAT GAT CAC CGG GGG CTT-3′; 18S rRNA forward: 5′-GCT TAA TTT GAC TCA ACA CGG GA-3′; 18S rRNA reverse: 5′-AGC TAT CAA TCT GTC AAT CCT GTC-3′. Measurement of 18S rRNA expression was used for normalization. Data were evaluated by the relative quantification method (2deltaCT). 

### 4.5. Immunoblotting

For quantitative immunoblotting, 50 µg of proteins from detergent extracts of TA muscles was subjected to SDS-PAGE and subsequent tank blotting, as previously described [19,46]. The nitrocellulose blot matrices were blocked at room temperature (RT), incubated with a polyclonal anti-nNOS antibody (N-7280, Sigma-Aldrich, Saint Louis, MO, USA) in a final 0.1 µg/mL concentration overnight at 4 °C and then developed by enhanced chemiluminescence (GE Healthcare, Glattbrugg, Switzerland). Ponceau S Red-staining of the blot matrices was conducted for normalization of the gel loading required for the densitometry. Therefore, the densitometric value for each nNOS-band at 160 kDa was divided by the densitometric values for the Ponceau S Red-stained protein bands (converted to black/white prior to densitometry) in the whole lane. If a clear disturbance of the Ponceau S Red coloration was visible anywhere on the nitrocellulose, only the densitometric densities of the homogeneously red-colored areas were included in the comparative quantitative analysis of all lanes of the blot matrix. The mean of the densities of the two samples from the group “control plasmid + sedentary” was set at 100% on each blot matrix.

### 4.6. Catalytic Histochemistry

NADPH diaphorase histochemistry specific to nNOS was carried out as previously reported [47]. Briefly, the sections were incubated in excess of the two substrates nitroblue tetrazolium salt (NBT) and NADPH in the presence of 2 M urea at RT, which reduced NBT to formazan at the sarcolemma specifically and proportionally to the nNOS concentration. The reaction was stopped after 30 min. Quantification of nNOS-specific diaphorase activity was performed by extraction of formazan from the tissue sections as previously described [47].

Alkaline phosphatase histochemistry on cross-sections of the TA muscle was carried out as previously described [48]. To determine the CF ratio as a structural index for capillarity, two randomly selected areas on both the oxidative and the glycolytic portions of the TA muscles, which were identified on serial sections by succinate dehydrogenase histochemistry [8], were photographed. The numbers of capillaries and muscle fibers were counted on the micrographs. 

### 4.7. Nitrate/Nitrite Fluorometric Assay

The nitrate/nitrite fluorometric assay (780051) from Cayman (Ann Arbor, MI, USA) was incubated with 50 minced TA muscle cryosections of 50 µm thickness for indirect measurement of NOS activity in tissue homogenates, as previously reported [9].

### 4.8. Fluorescence/Immunofluorescence Microscopy

For nNOS-immunohistochemistry, 10-µm thick cryosections of the TA muscle were subjected to fluorescence/immunofluorescence microscopy analysis as previously reported [46]. The cryosections were incubated with the polyclonal anti-nNOS antibody (N-7280, Sigma-Aldrich; diluted 1:20,000 with 5% BSA in PBS) for 2 h and then for 1 h with a goat anti-rabbit Cy5-conjugated secondary antibody (SAB4600045; Sigma-Aldrich, Saint Louis, MO, USA); diluted 1:500 with 5% BSA in PBS) at RT. Immunofluorescence images were recorded in a Zeiss 510 Meta-Laser scanning microscope (Axiovert 200 M, Lasers: HeNe (absorbance wavelength: 633 nm), HeNe (absorbance wavelength: 543 nm), Ar (absorbance wavelength: 488 nm) and were processed with IMARIS software, version 6.0.1 (Bitplane, Zurich, Switzerland). Densitometric quantification of nNOS-specific immunoreactivity in randomly selected individual muscle fibers was performed by image analysis on micrographs converted to black and white, as previously described [8]. nNOS expression on cryosections was calculated as the mean gray value of pixels at the sarcolemma related to anti-nNOS immunofluorescence signal per µm of sarcolemma length and expressed relative to the values in ZsGreen1-negative muscle fibers.

To determine the structural index ‘capillary contacts around skeletal muscle fibers’, a triple fluorescence microscopy analysis was carried out by combing ZsGreen1 fluorescence analysis to identify transfected TA muscle fibers with MHC-based fiber typing and reactivity of capillaries for isolectin B4 lectin from Bandeiraea (=griffonia) simplicifolia (BS-1 lectin). Because the autofluorescence of the reporter gene ZsGreen1 fainted and was only transiently detectable in our experiments, its expression was examined using a polyclonal anti-ZsGreen1 antibody (632474; Takara-Clontech, Saint-Germain-en-Laye, France). 

10 µm thick cryosections of the TA muscles were fixed in ice-cold acetone for 5 min. The dried sections were blocked in TBS, pH 7.4, supplemented with 5% BSA and 1 mM CaCl_2_ (TBS-BSA-Ca^2+^) for 30 min at RT and then incubated with the polyclonal anti-ZsGreen1 antibody (diluted to 1 µg/mL in TBS-BSA-Ca^2+^) overnight in a humid chamber at 4°C, washed 5 × 3 min in TBS-BSA-Ca^2+^ to be incubated with a secondary goat-anti-rabbit IgG Alexa Fluor 488-conjugated antibody (A-11008; Thermo Fisher, Darmstadt, Germany) diluted 1:2000 in TBS-BSA-Ca^2+^ for 1 h at RT. After 5 × 3 min washing in TBS-BSA-Ca^2+^ the sections were incubated with one of the three monoclonal anti-myosin heavy-chain (MHC) antibodies (SC-71 (MHC-IIA), 6H1 (MHC-IId/x) or BF-F3 (MHC-IIb); Developmental Studies Hybridoma Bank, Iowa City, IA, USA) in final concentrations of 0.1 µg/mL for 1 h in a humid chamber at RT. Then, 5 × 3 min washing in TBS-BSA-Ca^2+^ was followed by incubation with secondary goat-anti-mouse IgG Alexa Fluor 594-conjugated antibody (A-11032, Thermo Fisher) diluted 1:2000 in TBS-BSA-Ca^2+^ for 1 h at RT. An additional washing step (5 × 3 min with TBS-BSA-Ca^2+^) was followed by incubation with BS1-lectin conjugated to Alexa Fluor 405 (I21411; Thermo Fisher) in TBS-BSA-Ca^2+^ (1:1000) for 20 min at RT. Final washing in 5 × 3 min in TBS-BSA-Ca^2+^ was cover-slipped in glycerol and analyzed by fluorescence microscopy with a Zeiss Axioscope 40 microscope (Carl Zeiss, Oberkochen, Germany). Photomicrographs were digitally recorded and processed (Appendix A). 

### 4.9. Data Analysis and Statistics

All numerical data were expressed as mean values together with the standard deviation. All data sets were tested by Kolmogorov–Smirnoff for their normality of distribution prior to statistical analysis. Significance values were calculated by 2-tailed unpaired Student’s *t*-test for comparison of two study groups or a two-way analysis of variance (ANOVA) for multiple comparisons. If the outcome of the ANOVA was significant, implemented post hoc tests for multiple pairwise comparisons (Tukey) were performed. The significance level was set at *p* ≤ 0.05.

## Figures and Tables

**Figure 1 ijms-24-09341-f001:**
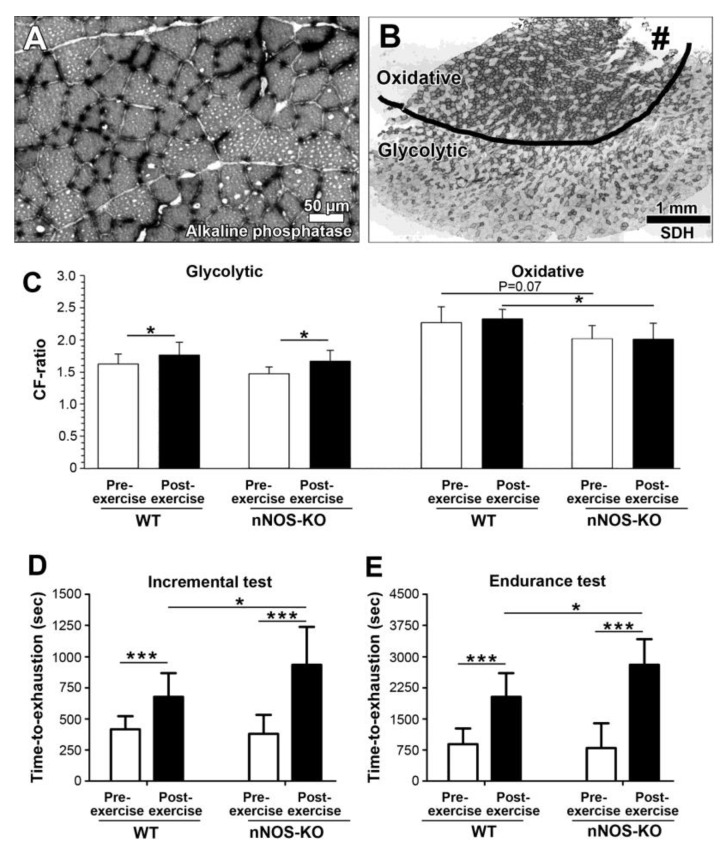
CF ratio in the tibialis anterior (TA) muscle and running performance of nNOS-KO mice and their WT littermates before and after 28 days of treadmill training. (**A**) A representative micrograph of a partial detail from a cross-sectioned TA muscle subjected to alkaline phosphatase histochemistry shows the capillary profiles as black dots that surround skeletal muscle fiber profiles of varying sizes. (**B**) Serial cross-sections of the TA muscles were subjected to succinate dehydrogenase (SDH) histochemistry to distinguish between the two TA muscle portions with different metabolic profiles: an oxidative part that appears darker due to many muscle fibers with high mitochondrial volume density and a much lighter glycolytic part in which most muscle fibers contain low volume densities of mitochondria. Note the torn area at the edge of the oxidative part (#) where the TA muscle was formerly attached to the tibia. (**C**) The CF ratio in both the glycolytic and the oxidative parts of the TA muscle was quantified in the trained nNOS-KO mice (*n* = 9) and their WT littermates (*n* = 8), as well as the sedentary mice of both strains (*n* = 4 each). (**D**,**E**) nNOS-KO mice (*n* = 9) and WT littermates (*n* = 8) were subjected to incremental (D) and endurance running tests (**E**) before and after the 28 days of treadmill training. Mean values ± SD, * or *** refer to values significantly differing (*p* ≤ 0.05 or *p* ≤ 0.001) between the groups applying a two-way ANOVA followed by paired post hoc Tukey’s multiple comparison test.

**Figure 2 ijms-24-09341-f002:**
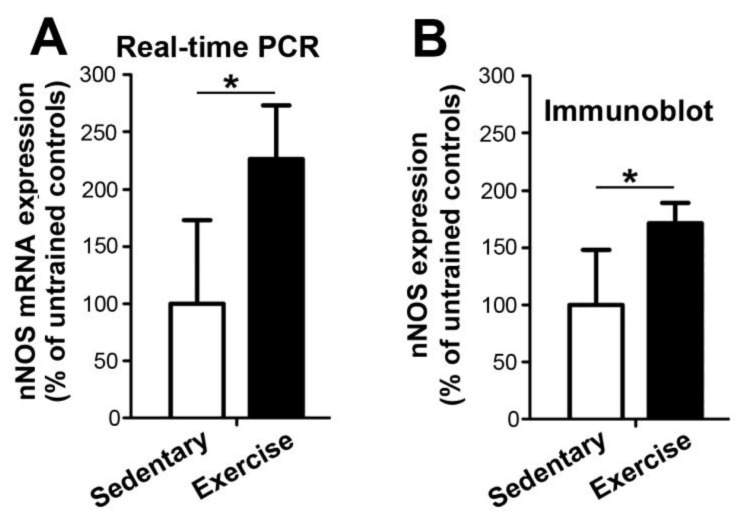
The mRNA (**A**) and protein (**B**) expression levels of nNOS in tibialis anterior (TA) muscle are higher in C57BL/6 mice undergoing treadmill training for 28 days than in littermates remaining sedentary. (**A**) Real-time PCR to quantify the mRNA expression levels of nNOS in TA muscle homogenates. The mean nNOS mRNA levels in the TA muscle of the sedentary mice were set as 100%. (**B**) The density of the nNOS bands reactive in immunoblots on TA muscle solubilsates was quantified. The mean nNOS protein levels in the TA muscle of the sedentary mice were set as 100%. *n* = 8 in both groups. Mean values ± SD, * refer to values significantly differing (*p* ≤ 0.05) between the two groups applying Student’s *t*-test.

**Figure 3 ijms-24-09341-f003:**
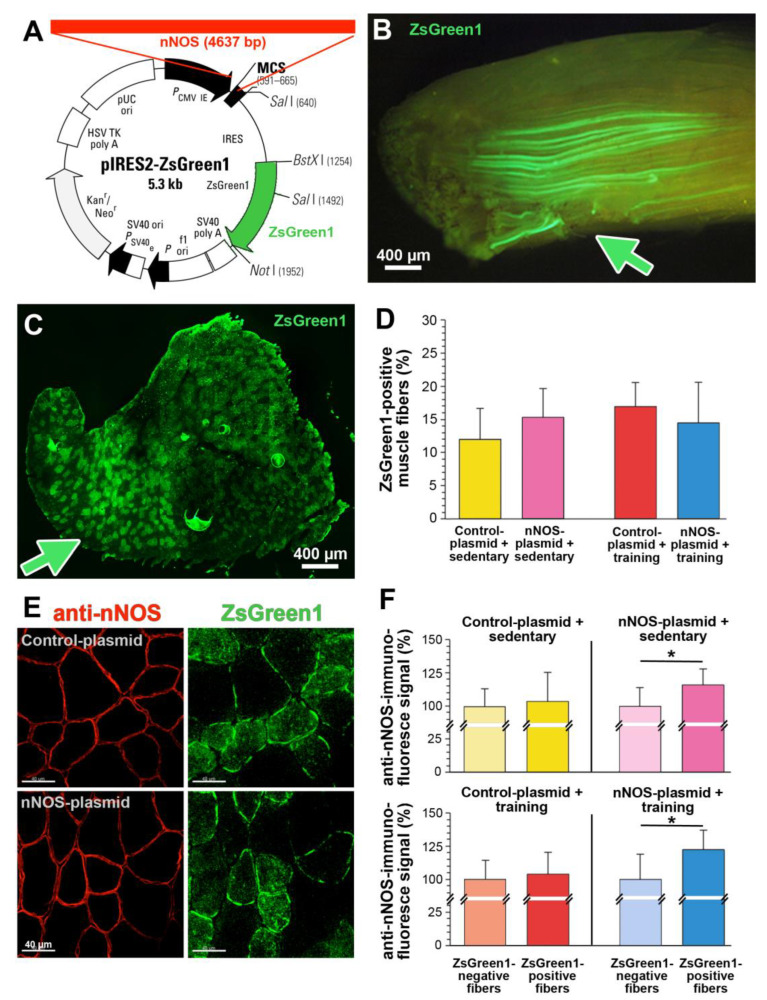
Gene electroporation of the nNOS plasmid and the plasmid without gene insert in the tibialis anterior (TA) muscle of C57BL/6 mice. (**A**) The expression plasmid pIRES2-ZsGreen1 containing the 4637 bp long nNOS gene was injected into the TA muscle of the left leg of 16 C57BL/6 mice. As a control for gene electroporation, the plasmid without a specific insert was injected into the TA muscle of the contralateral leg of each mouse. (**B**) The TA muscle surface of a sedentary mouse isolated eight days after gene electroporation with the pIRES2-ZsGreen1-nNOS plasmid was recorded by fluorescence stereomicroscopy. Skeletal muscle fibers taking up the plasmid were identified by the strong emission of green fluorescence produced by the reporter protein ZsGreen1. The green arrow labels one of the injection sites of the plasmid. (**C**) Overview of a cross-section of TA muscle from a sedentary mouse isolated eight days after gene electroporation with the reporter gene pIRES2-ZsGreen1-nNOS plasmid. The green arrow labels an injection site of the plasmid. (**D**) The number of skeletal muscle fibers expressing ZsGreen1 was set in relation to the number of all muscle fibers on the TA muscle cross-sections. *n* = 3 in all groups. Mean values ± SD. (**E**) Two examples of cross-sectioned TA muscle subjected to anti-nNOS-immunohistochemistry (red fluorescence; 594 nm), which were isolated from a sedentary mouse eight days after gene electroporation with the pIRES2-ZsGreen1 plasmid (left TA) or the pIRES2-ZsGreen1-nNOS plasmid (right TA). Muscle fibers that had incorporated the plasmid were identified by the green fluorescence emission (493 nm) evoked by ZsGreen expression. (**F**). Densitometric quantification of the nNOS-immunohistochemical reactivity in ZsGreen1-negative and ZsGreen1-positive muscle fibers within TA muscle of sedentary or trained mice isolated eight days after gene electroporation with either the control plasmid or the nNOS-gene containing plasmid. *n* = 6–10 randomly selected ZsGreen1-negative and ZsGreen1-positive muscle fibers from each of six TA muscles in each of the four study groups. The mean value of nNOS immunoreactivity in the ZsGreen1-negative muscle fibers on each section was set as 100%. Mean values ± SD, * refer to values significantly differing (*p* ≤ 0.05) between ZsGreen1-negative and ZsGreen1-positive in Student’s *t*-test.

**Figure 4 ijms-24-09341-f004:**
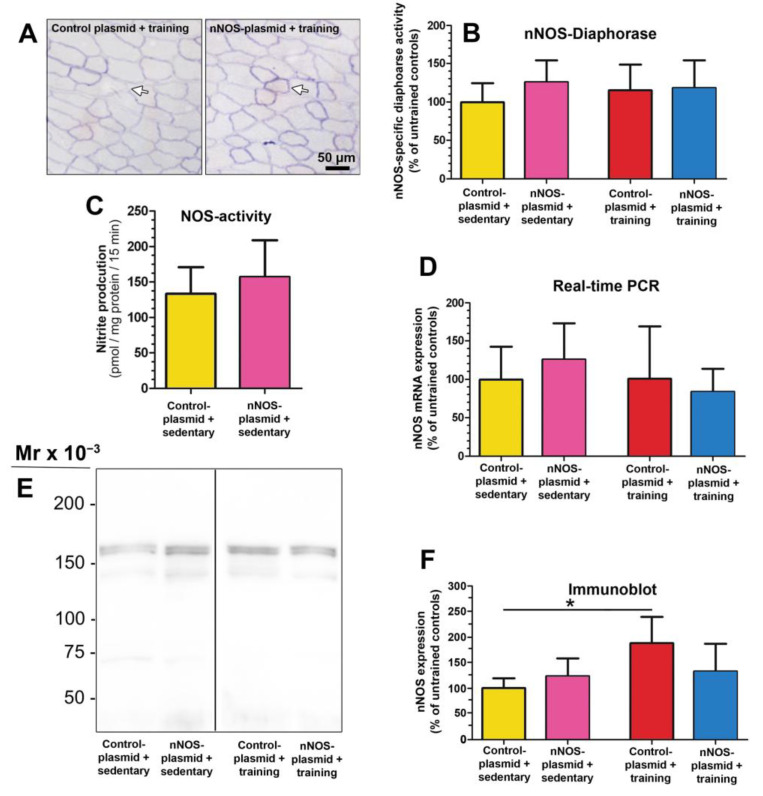
Total nNOS diaphorase activity, NOS activity, nNOS mRNA and protein levels did not differ between the tibialis anterior (TA) muscle transfected with the nNOS plasmid and those transfected with the control plasmid (without/with treadmill training). (**A**) Isolated at Day 8 after the gene electroporation, nNOS-specific diaphorase activity was restricted to the sarcolemma on cross-sections of all TA muscles (arrows). (**B**) Densitometric quantification of nNOS-specific diaphorase activity demonstrated similar formazan production rates in the cross-sections of TA muscle transfected with the control plasmid or the nNOS plasmid of mice that remained sedentary (*n*= 6 for both plasmids) or underwent treadmill training (*n*= 6 for both plasmids). (**C**) The nitrite production rates, which correspond to NOS activity, were assessed by a fluorometric assay based on the nitrosylation of 2,3-diaminonaphthalene (DAN) to yield fluorescent 2,3-naphthotriazole in homogenates of TA muscle transfected either with the control plasmid or the nNOS plasmid from sedentary mice (*n*= 6 for both plasmids). Mean values ± SD, * refer to values significantly differing (*p* ≤ 0.05) between the groups in Student’s *t*-test. (**D**) By Real-time PCR, the mRNA levels of nNOS were assessed in the TA muscle of the mice subjected to gene electroporation. Only non-significant differences in nNOS mRNA levels were gauged in homogenates of TA muscle transfected with either the control plasmid or the nNOS plasmid of the mice that remained sedentary or underwent treadmill training for 7 days (*n*= 7–8 mice in each of the four groups). The nNOS mRNA levels in the TA muscle of the control plasmid-transfected, untrained mice were set as 100%. (**E**) nNOS expression levels at the protein level were evaluated by quantitative immunoblotting in solubilsates of TA muscles transfected with either the control plasmid or the nNOS plasmid from mice remaining sedentary or undergoing treadmill training for seven days. (**F**) The nNOS-immunoreactive bands in the TA muscle solubilsates of the mice from the four study groups (*n*= 6 mice each) were densitometrically quantified and normalized to protein loading assessed by Ponceau S-staining. The nNOS protein levels in the TA muscle of untrained, control plasmid-transfected mice were set as 100%. Mean values ± SD, in (**D**,**F**) * refer to values of treated mice significantly differing (*p* ≤ 0.05) plasmid-transfected sedentary group in a two-way ANOVA followed by paired post hoc Tukey’s multiple comparison test.

**Figure 5 ijms-24-09341-f005:**
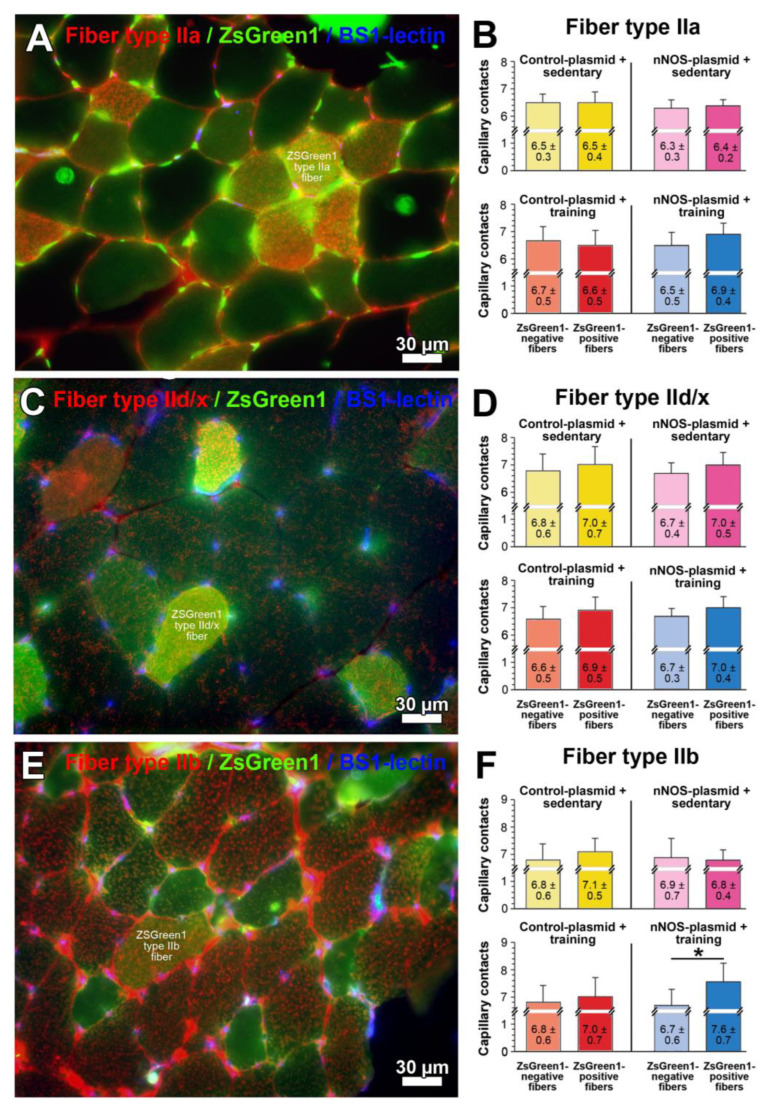
Quantification of the fiber-type-specific capillarity in the tibialis anterior (TA) muscle of mice included in the gene electroporation study. (**A**,**C**,**E**) Cryosection of the transfected TA muscles was subjected to immunohistochemistry with anti-ZsGreen1 (transfected fibers; green) and anti-myosin heavy-chain (MHC) antibodies (skeletal muscle fiber types as indicated; red) followed by histochemistry with Bandeiraea simplicifolia (BS)1-lectin specific for capillaries. Shown are representative micrographs of MHC-IIa (**A**), MHC-IId/x (**C**) and MHC-IIb (**E**) immunoreactive muscle fibers in a plasmid-transfected TA muscle of a sedentary mouse as example after merging of the images. (**B**,**D**,**F**) The number of capillary contacts as index for capillarity was determined on 4–10 randomly selected ZsGreen1-negative and ZsGreen1-positive skeletal muscle fibers from six TA muscles in each of the four study groups. The mean value for capillary contacts around ZsGreen1-negative muscle fibers was set as 100% for each muscle fiber type. Mean values ± SD, * refer to values significantly differing (*p* ≤ 0.05) between ZsGreen1-negative and ZsGreen1-positive in Student’s *t*-test.

## Data Availability

Data supporting reported results are available from the corresponding authors to all interested researchers.

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
