# Peer review of "nNOS Increases Fiber Type-Specific Angiogenesis in Skeletal Muscle of Mice in Response to Endurance Exercise"

_ijms, 2023, doi:10.3390/ijms24119341_

Round 1

Reviewer 1 Report

Dear Authors,

Manuscript "nNOS increases fibre type-specific angiogenesis in skeletal muscle of mice in response to endurance exercise." deal with a complex and multi-level aspect of NOS-regulated angiogenesis in muscle tissue. The authors decided to examine an in vivo working model, which is challenging and requires multiple complicated research techniques. 

The research design and its conduct and result description are proper. 

There are only minor remarks I noticed, which I listed below:

Line 89, which is the most serious question from my side: 

"angiogenesis occurred specifically in the glycolytic portion  of the [...]." 

that suggests another critical player of angiogenesis-that is, in this case, intermittent hypoxia. Literature shows that hypoxia is a potent angiogenesis regulator, and therefore in presented research model would have a profound impact. My question is, why do authors exclude this aspect from their research agenda?

Fig 4.e 

In many research papers, it is suggested to attach an original picture of the blot for better quality. The same applies to Ponceau S staining. 

Real time PCR:

Was reaction efficiency of control equal to nNOS?

WB:

Could you please describe the necessary details of normalisation with Ponceau-S staining.?

 Fluorescence:

Could you please give all details on temperatures with antibody incubations?

Kind regards,

Author Response

Reviewer 1

Manuscript "nNOS increases fibre type-specific angiogenesis in skeletal muscle of mice in response to endurance exercise" deal with a complex and multi-level aspect of NOS-regulated angiogenesis in muscle tissue. The authors decided to examine an in vivo working model, which is challenging and requires multiple complicated research techniques. The research design and its conduct and result description are proper.

We would like to thank the reviewer for her/his positive review and the constructive feedback.

There are only minor remarks I noticed, which I listed below:

Line 89, which is the most serious question from my side:

"angiogenesis occurred specifically in the glycolytic portion of the [...]." that suggests another critical player of angiogenesis-that is, in this case, intermittent hypoxia. Literature shows that hypoxia is a potent angiogenesis regulator, and therefore in presented research model would have a profound impact. My question is, why do authors exclude this aspect from their research agenda?

It is indeed an interesting and important objection raised by the reviewer. In interpreting our findings that angiogenesis was primarily realized in the glycolytic IIb fibers, we speculated that these fibers are particularly susceptible to the increased contractility caused by treadmill training (lines 356 and 380). As the reviewer points out, it could actually be that the growth of the capillary system was also induced by training-related hypoxia in the skeletal muscles. For example, smooth muscle cells are known to change nNOS expression in a hypoxia-sensitive manner (Ward et al, 2005; PMID: 16276418). We have now expanded the discussion to include this point.

Fig 4e: In many research papers, it is suggested to attach an original picture of the blot for better quality. The same applies to Ponceau S staining.

We agree with the reviewer that showing images of the original immunoblots and the subsequent Ponceau S Red stainings of the blot matrices would be helpful in enhancing the quality of the manuscript. But instead of just adding a selected Ponceau S Red stainings in the Fig. 4e, we included the blot matrices of all samples that were generated and evaluated in our study in their original state as Supplementary Figure.

Real-time PCR: Was reaction efficiency of control equal to nNOS?

Unfortunately, we did not determine the efficiency of the Real-time PCR using a standard curve with nNOS cDNA during experimentation. However, we can confirm that the amplification curves of the real-time PCR were more or less congruent (slopes of the amplification curves in the exponential phase intersecting the SybrGreen threshold line were parallel), so we assumed that the determinations of the CTs were suitable to determine the nNOS mRNA concentrations.

WB: Could you please describe the necessary details of normalisation with Ponceau-S staining?

We extended the description of the normalization process with Ponceau S Red staining, as requested by the reviewer.

Fluorescence: Could you please give all details on temperatures with antibody incubations?

The incubation temperatures are now specified, as requested by the reviewer. Sorry for the negligence.

Reviewer 2 Report

Oliver Baum et al. found that exercise can induce nNOS in type-IIb muscle fibers and demonstrated the role of nNOS in angiogenesis. Their findings suggested that nNOS is a crucial molecule linking exercise and angiogenesis, thereby shedding light on a new aspect of the physiological function of nNOS. The authors used mice muscles transfected (electroporation) with nNOS expression vector and examined the muscle fibers immunohistochemically and biochemically. Since the authors investigated the results both in plasmid -/+ and in training -/+ groups, respectively, their experimental design seems sound, and they conducted the experiments well. They also provided sufficient pieces of evidence to draw their conclusions. The reviewer will recommend publishing this manuscript in IJMS.

Minor points

Because of the lack of regulatory sequence in the introduced IRES plasmid, the exogenously expressed nNOS might be produced constitutively. However, the result demonstrated fiber type-specific nNOS expression. Furthermore, the reviewer wonders how treadmill training can affect expression from the insert of the plasmid.

In Fig.4 D, F, the expression level of nNOS is higher in “control plasmid + training” than in “nNOS plasmid + training.” 

If the level of nNOS in “nNOS plasmid + training” is not different from that of “control plasmid + sedentary,” the authors could not demonstrate the effect of introduced nNOS.

In the living conditions, mice are not always sedentary. Therefore, capillary contacts are more prominent around type-IIb fibers in the basal condition, given that nNOS exerts its effects type-specific manner. It will be informative if the authors provide such data.

The authors described the term “CF-ratio” without explanation, but the term appears not to be familiar to readers. 

Author Response

Reviewer 2

Oliver Baum et al. found that exercise can induce nNOS in type-IIb muscle fibers and demonstrated the role of nNOS in angiogenesis. Their findings suggested that nNOS is a crucial molecule linking exercise and angiogenesis, thereby shedding light on a new aspect of the physiological function of nNOS. The authors used mice muscles transfected (electroporation) with nNOS expression vector and examined the muscle fibers immunohistochemically and biochemically. Since the authors investigated the results both in plasmid -/+ and in training -/+ groups, respectively, their experimental design seems sound, and they conducted the experiments well. They also provided sufficient pieces of evidence to draw their conclusions. The reviewer will recommend publishing this manuscript in IJMS.

We would like to thank the reviewer for her/his positive review and the constructive feedback.

Minor points

  1. Because of the lack of regulatory sequence in the introduced IRES plasmid, the exogenously expressed nNOS might be produced constitutively. However, the result demonstrated fiber type-specific nNOS expression. Furthermore, the reviewer wonders how treadmill training can affect expression from the insert of the plasmid.

The reviewer's question is very valid as to how it should be possible to measure a cellular effect of nNOS on treadmill training if the nNOS gene and plasmid inserted by electroporation do not exhibit promoter sequences for regulated nNOS expression. However, this question does not take into account that skeletal muscle fibers still carry the intrinsic nNOS gene with regulatory sequences, giving these fibers the potential to induce nNOS expression in response to endurance exercise (Figure 2). Accordingly, we were able to examine skeletal muscles with four different nNOS expression patterns (corresponds to the four study groups of mice): 1. intrinsic nNOS from sedentary muscle; 2. intrinsic nNOS + transfected nNOS from sedentary muscle; 3. intrinsic nNOS from trained muscle and 4. intrinsic nNOS + transfected nNOS from trained muscle. We have now emphasized this study design more in the Discussion.

  1. In Fig.4 D, F, the expression level of nNOS is higher in “control plasmid + training” than in “nNOS plasmid + training.” If the level of nNOS in “nNOS plasmid + training” is not different from that of “control plasmid + sedentary,” the authors could not demonstrate the effect of introduced nNOS.

The reviewer correctly summarized the main conclusion we reached after evaluating nNOS transfection in whole muscle homogenates. We could not demonstrate a significant increase in nNOS expression using any experimental method when using muscle cryosections (Figure 4A-C) or homogenates as starting material (Figure 4D,E). Consequently, we then restricted the analysis of whether nNOS was up-regulated by the gene electroporation procedure (Figure 3E,F) and has an effect on capillarity (Figure 5) to the muscle fiber level. This background to our decision to carry out the capillarity analyzes at muscle fiber level is now mentioned in the Results section.

  1. In the living conditions, mice are not always sedentary. Therefore, capillary contacts are more prominent around type-IIb fibers in the basal condition, given that nNOS exerts its effects type-specific manner. It will be informative if the authors provide such data.

Under basal conditions, nNOS shows a fiber type-specific expression pattern in rodent skeletal muscle: I<IIb<IIa≈IId/x (Planitzer et al., 2001; PMID: 11702244). Under basal conditions, the capillary contacts (which does not consider the mean cross-sectional fiber area) in rodent skeletal muscle are fiber-type-specifically distributed: I≈IIa>IId/x≈IIb (Waters et al., 2004; PMID: 15253894). These semi-quantitative data suggest that there is no correlation between nNOS expression and capillarity under basal condition. In contrast, nNOS expression and capillarity simultaneously increase in response to endurance training in mice, especially in IIb muscle fibers, as referred to in the introduction. The data from the study here show that nNOS actually has an upstream impact on the growth of the capillary network in IIb muscle fibers. Accordingly, nNOS does not affect basal mitochondrial density but rather mitochondriogenesis during endurance training (Baum et al., 2018; PMID: 29678764).

  1. The authors described the term “CF-ratio” without explanation, but the term appears not to be familiar to readers.

The numerical capillary-to-fiber ratio (CF-ratio) represents the number of capillaries divided by the number of skeletal muscle fibers within a given area. This index should not be confused with the capillary density (CD), which is the number of capillary profiles per cross-sectional area of muscle fibers. In contrast to the CF-ratio the CD depends on the mean cross-sectional area of muscle fibers, which is often modulated in response to changes in physical demand or the metabolic environment. If the CF-ratio in skeletal muscle changes over time, angiogenesis has occurred and the magnitude of the change in CF ratio provides information about the extent of the angiogenic growth of the capillary system. For an overview of the determination and functional relevance of the structural indices for capillarization in skeletal muscle, we recommend the article by Olfert et al. (2016; PMID: 26608338), which we also referred to in the introduction.

The index CF-ratio is now defined, as requested by the reviewer. Sorry for the negligence of omitting this definition.